# Prevalence of Possible Sleep Bruxism and Its Association with Social and Orofacial Factors in Preschool Population

**DOI:** 10.3390/healthcare11101450

**Published:** 2023-05-16

**Authors:** Montserrat Diéguez-Pérez, Jesús Miguel Ticona-Flores, Beatriz Prieto-Regueiro

**Affiliations:** 1Department of Preclinical Dentistry, Faculty of Biomedical and Health Sciences, Universidad Europea de Madrid, Villaviciosa de Odón, 28670 Madrid, Spain; 2Faculty of Biomedical and Health Sciences, Universidad Europea de Madrid, Villaviciosa de Odón, 28670 Madrid, Spain; jesus.ticona.f@upch.pe (J.M.T.-F.); breica68@hotmail.com (B.P.-R.)

**Keywords:** sleep bruxism, bruxism, childhood sleep bruxism, nocturnal bruxism, child, preschool

## Abstract

The prevalence of bruxism in the preschool population varies according to different investigations. The aim of this study was to investigate the prevalence of sleep bruxism and its relationship with social and orofacial factors in children aged 3 to 5 years. Three hundred forty-three preschool children were divided into two groups, one with nocturnal bruxism, as reported by parents, and another without this disorder. Questionnaires were distributed to the parents to determine the children’s family social status and parafunctional habits at the time of the study. The clinical inspection confirmed the presence of certain parafunctional habits and the children’s occlusal characteristics. The data obtained were analysed using descriptive statistics such as frequency chi-square tests to identify the influence of qualitative variables. Of the total sample, 28.9% of preschoolers presented sleep bruxism. The highest prevalence was observed in boys (61.6%) at the age of 5 years (41.4%). Characteristics associated with this pathology were lip incompetence, open bite, crossbite, and overbite, with *p* < 0.05. Sleep bruxism in preschool children has a higher prevalence in boys and is more frequently expressed from 5 years of age. Open bite, overbite, and crossbite should be considered factors associated with parafunction.

## 1. Introduction

Bruxism is a non-functional and repetitive involuntary activity of the masticatory muscles [1]. Its prevalence in the preschool population varies according to different investigations because it is difficult to diagnose this parafunction at these ages [2,3,4]. Some genetic polymorphisms are associated with this habit. This contributes to the multifactorial aetiology of this condition in children [5]. In the paediatric population, this manifestation is nocturnal and, as it is associated with circadian rhythm, it is called sleep bruxism. If left untreated for an extended period, it can affect the stomatognathic system, favouring the appearance of masticatory dysfunctions. In other cases, these dysfunctions favour the appearance of bruxism [1]. In the long term, they can cause temporomandibular disorders [6,7,8], are responsible for headaches and orofacial pain associated with waking up [9,10,11,12], produce sleep disturbances such as nightmares and snoring, and cause respiratory difficulties and variations in heart rate, among other manifestations [13,14,15,16,17,18,19]. Furthermore, children with bruxism are more predisposed to experiencing stress and behavioural problems associated with nocturnal habits [20,21]. The non-application of therapies in childhood contributes to worsening consequences in adulthood [22].

Paediatricians must be trained to issue presumptive diagnoses in children and educate parents to prevent complications that affect oral and systemic health [1]. The three levels of diagnosis described by Lobbezoo et al. are based on the information the parents or relatives transmit to the professional regarding the dental noise that the child produces during sleep; in this case, it is a possible diagnosis [23]. Clinical manifestations compatible with bruxism, such as wear and tear, discomfort, masticatory muscle hypertrophy, headaches, and lesions in oral mucosal tissues, constitute a probable diagnosis. Only with polysomnography is a definitive diagnosis possible [1]. Given the difficulty of diagnosis, a multidisciplinary approach is essential [3], as is identifying risk factors, because these will influence the clinical management strategy [14].

The therapeutic management of child bruxism at the dental level is complicated. Some therapeutic options include kinesitherapy, massage, infrared, and low-intensity laser therapy [1]. The use of oral devices can inhibit maxillary bone growth. For this reason, some clinicians opt for therapies based on neuro-occlusal rehabilitation [24].

Drug treatments with hydroxyzine, trazodone, flurazepam, occlusal splints, and orthodontic and psychological therapies are therapeutic options for adults [4].

The factors predisposing individuals to this parafunction are still a topic of debate; because of this, it is essential to develop research in this field. Thus, the main aim was to determine the prevalence of sleep bruxism and its relationship with family social class, parafunctional habits, and occlusal characteristics in the preschool population as possible factors associated with bruxism.

## 2. Materials and Methods

A descriptive, observational, cross-sectional, and epidemiological study was designed. It was carried out under the statutes and approval of the Ethics Committee of the European University of Madrid and the Clinical Research Ethics Committee of the Insular-Maternal and Child University Hospital Complex (CEIC-CHUIMI-2015/807).

### 2.1. Geographical Area, Sample Size and Selection Criteria

Data collection was performed before the declaration of a state of alarm due to coronavirus in the municipality of Arrecife (Lanzarote, Canary Islands).

The sample size was calculated considering a total population of 2108 users, an expected proportion of 30% [10], a confidence level of 95%, a precision of 5%, and a power of 80%, with a resulting 283 patients determined as the minimum number needed.

A significance level of *p* < 0.05 was determined for the minimum number of 283 patients with a power of 80%. Meanwhile, 343 patients in total were chosen for the final sample.

This study included healthy patients aged 3 to 5 years with a Canarian health card who attended paediatric consultations in Arrecife and whose parents signed the consent form allowing them to participate in all study phases. Patients without personal and complete medical data, negative behaviour (Frankl scale), and mixed dentition were excluded (Figure 1).

### 2.2. Questionary 

The lead researcher in the primary care paediatric office performed the data collection three days a week with a simple random sampling technique.

After obtaining the informed consent and the affiliation and medical data, the questionnaire was completed via a guided interview with the parents, where information was collected on breastfeeding and parafunctional and non-nutritive habits such as time of use of the bottle and pacifier, presence of the digital sucking habit, onychophagia, mouth breathing, or lip incompetence. 

The diagnosis of bruxism was based on the information reported by the parents. Finally, the family’s social status data were collected (Table 1), considering the head of the household occupation and the person who regularly contributed the most to the household budget [25].

The average time spent on the interview was approximately 5 min per participant.

### 2.3. Clinical Exam 

The recommended methodology by the WHO was followed. A previously trained and calibrated researcher conducted the examination in the dental chair (Fedesa JS 500 Spanish manufacturing) with the preschoolers in semi-fowler position and the neck in extension. The intraoral examination was performed with a mouth mirror without magnification while the examiner stood [26].

The presence of habits such as oral breathing was confirmed by observing the lack of lip seal and the presence of adenoid facies and lower lip incompetence. Finger sucking and onychophagia were verified via visual inspection of the child’s fingers.

The presence of primary teeth and interdental spaces were recorded, along with the occlusal characteristics in the three anatomical planes (Figure 2). In the Sagittal plane: the presence of overjet and the occlusal relationship of the primary molars were assessed, with the latter determining the future occlusal relationship of permanent molars. The following criteria were considered: (1) Flush terminal plane; (2) short mesial step; (3) long mesial step; (4) distal step; (5) mixed component when a favourable occlusion coexisted on one side of the dental arch and malocclusion on the other. The same occlusal characteristics did not exist on both sides of the arch [27]. In the vertical plane, overbite and anterior open bite were recorded. In the transverse plane, the presence or absence of crossbite was described.

It was considered that an occlusal alteration was present when the child exhibited an overjet, a long mesial step, a distal step, and a mixed component, as well as an overbite of 2/3 and 3/3, open bite, and crossbite. 

The time used in the examination per patient was approximately 10 min.

To evaluate intra-observer reliability, a second clinical examination was conducted on 10% of the sample one week after the first inspection. The obtained kappa value was 0.9, indicating a significantly high level of agreement.

### 2.4. Statistical Analysis 

SPSS IBM version 26.0 for Windows was utilised for the statistical analyses. Descriptive statistics, such as frequencies, were employed to represent the qualitative variables of the sample. Contingency tables and the chi-square test were utilised for inferential statistics to assess the independence or influence between two qualitative variables. Furthermore, the prevalence ratio was employed to measure the association, 95% confidence intervals were calculated, and a significance level of *p* < 0.05 was established.

## 3. Results

### 3.1. Sample Description 

The initial population consisted of 361 patients. However, the final sample size for this study was 343, as participants were excluded due to non-attendance of the medical appointment, and thirteen were excluded due to lack of cooperation from the preschool. Of all the preschoolers included, 99 (28.9%) were reported by their parents as having bruxism. The distribution of the sample is presented in Figure 3. Among the preschoolers with bruxism, 61 (61.6%) were boys, while 38 (38.4%) were girls. In contrast, among the preschoolers without parafunction, 114 (46.7%) were boys, and 130 (53.3%) were girls. Regarding age distribution within the bruxism group, 18 (18.2%) children were three years old, 40 (40.4%) were four years old, and 41 (41.1%) were five years old. In comparison, among the children without the habit, 81 were three years old, 82 (33.6%) were four, and 81 (33.3%) were five. The application of the chi-square test revealed statistically significant differences in terms of gender (*p* = 0.017) and preschool age (*p* = 0.05).

### 3.2. Social Class 

Among the parents of the total number of preschoolers, 29.7% were semi-skilled manual workers, the distribution of the presence or absence of bruxism in each family social class is presented in Table 2, and the comparison between each social class is depicted in Figure 4.

### 3.3. Parafunctional Habits 

When analysing the association between registered parafunctional habits and possible bruxism in preschoolers, significance was only found in the variable of lip incompetence, as shown in Table 3. The comparison of risk factors is illustrated in Figure 5.

### 3.4. Occlusal Characteristics 

Several occlusal factors are significantly associated with the habit of bruxism, such as occlusal alterations in the vertical and transverse planes. These factors are presented in Table 4 and visualised in Figure 6. 

## 4. Discussion

The diagnosis of sleep bruxism in young children is commonly based on the report of family members seeking advice due to the annoying grinding sound. Access to polysomnographic records is limited, and clinical manifestations of bruxism are not as evident as in adults, particularly children aged 3–5. Therefore, most studies in this age group have been conducted based on possible diagnoses. This underscores the need for further investigation to enhance our knowledge of the condition in this population and develop effective diagnostic and therapeutic strategies [28,29]

### 4.1. Bruxism Prevalence 

The prevalence of sleep bruxism in early childhood shows great variability according to the scientific literature. Bulanda et al. [1] determined a 13–49% prevalence rate. However, these authors did not refer specifically to children aged 3–5 years; they referred, in general, to children without indicating specific ages. This variability may be because of the different age groups evaluated and the heterogeneity of the diagnostic methods. In this regard, this study aimed to address the dearth of data on preschool children.

According to Clementino et al. [28], the habit of bruxism in children aged 3, 4, and 5 years is practically non-existent (2.0%, 15.1%, and 1%, respectively). This is the only study in the scientific literature determining the prevalence with the same age ranges as our research (5.2%, 11.6%, and 11.7%, respectively). The high prevalence observed in our results may derive from the ignorance of the multifactorial aetiology of bruxism and the modern lifestyle.

Other studies, such as Ghafournia et al. [29], observed a prevalence of 12.7% in preschool children aged 3–6 years, which is lower than our results. The presence of probable bruxism in children aged 2–5 years, according to Massignan et al. [30], was 22.3%. In contrast to this study, the lower prevalence may be derived from the inclusion of 2-year-old participants. Gomes et al. [31] indicated that 26.9% of 5-year-old children have this habit, which is again a lower prevalence than ours, as indicated by Nahás-Scocate et al. [12] in children aged 2–6 years (26.5%). In these investigations, in addition to the values found being lower than this research, the prevalence does not correlate with the increase in the child’s age.

The results of other researchers indicate a higher prevalence. According to Alencar et al. [13], 32.8% of the 3 to 7 years old sample present this habit. However, it must be considered that the age ranges studied in this case also refer to older children. Insana et al. [32] observe how 36.8% of preschool children are bruxists. Ramos et al. [33] indicate a possible prevalence of bruxism in 5-year-old preschoolers at 36%. The results of Da Costa et al. [34] go even further, as they report that 47.4% of children between 4 and 5 years old have this non-functional habit, although it is true that they have not considered the age of 3 years. In contrast, Simões-Zenar et al. [19] observed a much higher prevalence (55.3%) of bruxism reported by parents in children between 4 and 6 years of age, but they included 6-year-old children. 

### 4.2. Association with Sex 

In this regard, a consensus has yet to be reached. Alves et al. [22] reported a prevalence of 25.2% for bruxism in the age range of 3–12 years, with a significant association found with the female gender (*p* = 0.034). Similarly, Clementino et al. [28] identified a frequency of 64.4% in the same age range, with a preference for the female sex (*p* = 0.020). However, this study found a statistically significant difference, with the male group being the most prevalent, possibly due to unconsidered factors such as personality traits or anxiety [35], as well as the inclusion of preschoolers and a lower age range in the analysis. On the other hand, Da Costa et al. [34] did not find significant differences based on sex. Nevertheless, they observed a higher prevalence of bruxism in boys. Ghafournia et al. [29], Simões-Zenar et al. [19], Gomes et al. [31], Massignan et al. [30], and Nahás-Scocate et al. [12] also failed to establish a statistically significant relationship between gender and bruxism (*p* = 0.097; *p* = 0.393). Similarly, De Alencar et al. [35] and Ferrari-Piloni [36] did not observe statistically significant differences regarding sex in the 3–7 years age range.

### 4.3. Association with Social Factors 

Similar to the sex variable, there is no consensus on social factors. Perhaps the age of the children studied may influence this aspect, as their emotional development can influence managing family dynamics derived from the social aspect. Some researchers suggest a significant relationship between high and medium socioeconomic status [37]. However, other authors [30,31] found no significant association between families’ social class and sleep bruxism. On the other hand, it has also been stated that the upper-middle class exhibit the highest prevalence of bruxism [35]. In this study, it has been observed that when the parents were skilled manual workers, the prevalence of bruxism was higher than when the parents had other occupations. This could be due to the more extreme working hours of these professionals, which may impact family relationships and consequently promote the presence of this habit.

In this context, parents’ social status may serve as a proxy for various bruxism-related factors, including lifestyle factors resulting from occupation, work engagement, and psychological traits. An in-depth understanding of these potential factors may facilitate a better comprehension of the aetiology of bruxism in children. It may contribute to the development of effective prevention and intervention strategies.

### 4.4. Association with Orofacial Factors and Habits 

The highest prevalence in our sample was associated with bottle use ≥24 months (51.5%) and pacifier use <24 months (62.6%). DaCosta et al. [34] found significance in patients with onychophagia, lip biting, gum chewers, and mouth breathing. Gomes et al. [31] also found significance in preschoolers with onychophagia (*p* = 0.112). Simões-Zenari et al. [19] observed significant values in their study of 4 to 6-year-old school children when relating bruxism to pacifier use (*p* = 0.036), lip biting (*p* < 0.001) and onychophagia (*p* = 0.028). Regarding pacifier use, children with this habit are seven times more likely to have bruxism. Children who bite their lips are five times more likely to be bruxists. In the present research, the presence of onychophagia (27.3%) and oral breathing (14.1%) and their relationship with sleep bruxism were insignificant. Research with populations of higher age ranges (8 to 10 years) indicates that children with sleep bruxism more frequently have a history of onychophagia (62.3%), lip sucking (63.2%), handkerchiefs (48.4%), and fingers (63%). However, these results are not statistically significant except for sucking on objects (*p* = 0.001) [38]. Lamenha Lins et al. [39] reported a significant association between oral breathing and sleep bruxism in children (*p* < 0.001), who are 2.71 times more likely to be bruxists.

Emodi-Perlman et al. have reported that habits such as pacifier use and onychophagia increase the likelihood of a child developing sleep bruxism. Therefore, healthcare professionals are advised to be vigilant in identifying clinical signs of bruxism in children who exhibit or have a history of pacifier use and nail-biting [40]. These habits have been suggested to increase the risk of bruxism due to repetitive oral movements and increased muscle activity in the jaw and face. Therefore, identifying these habits may aid in the early detection and management of sleep bruxism in children. However, further research is needed to elucidate the precise mechanisms underlying the association between these habits and sleep bruxism.

### 4.5. Association with Occlusal Characteristics 

According to Da Costa et al. [34], children with class I canine relationships and significant overbites have a higher likelihood of experiencing bruxism. This study supports their findings as the most prevalent occlusal characteristics align with the results of these authors. The 2/3 overbite (34.3%) and flush terminal plane (53.5%) are commonly observed in individuals with bruxism. However, no statistically significant differences were found regarding the occlusal relationship in the sagittal plane. Ghafournia et al. [29] identified statistically significant associations between bruxism and the flush terminal plane (*p* = 0.023) as well as the mesial step (*p* = 0.001) in preschool children aged 3–6 years. Considering the discrepancies between these investigations, it is necessary to consider the peculiarities of masticatory function rather than solely focusing on occlusal characteristics since an abnormal mandibular function can lead to bruxism regardless of the child’s occlusion type. In this present study, bruxist preschoolers exhibit crossbite (17.2%) and open bite (5.1%), which are statistically significant occlusal characteristics, unlike other investigations [29]. Nahás-Scocate et al. [12] also found significance in the relationship between bruxism to posterior crossbite (*p* = 0.002). Although, according to Gomes et al. [31], malocclusion is not significantly associated with sleep bruxism (*p* = 0.226). Once again, this reinforces the idea that the masticatory function is relevant.

### 4.6. Association with Diastemas and Dental Crowding 

There has not been found research that indicates the presence or absence of diastemas as a factor associated with sleep bruxism. Although this characteristic indicates the degree of maxillary bone development based on the observed results, it is not a statistically significant factor concerning bruxism. (*p* = 0.656). In a similar line of research, Hazar Bodrumlu et al. [40] found that the length and width of the maxillary arch in child patients are not associated with bruxism. However, the scientific literature reports an association between bruxism and the presence of dental crowding [41].

### 4.7. Strengths and Limitations 

Using questionnaires to diagnose bruxism in preschool children is a reliable and cost-effective method. However, parents sometimes need to be made aware of parafunctional behaviours, partially because children sleep separately from their parents, which could lead to underestimating the prevalence using this diagnostic method. In addition, at these ages, the clinical manifestations related to bruxism are mild and can easily go unnoticed.

Paediatric sleep bruxism serves as an indicator of adverse health status and highlights the need for early intervention. Multiple etiological factors in a child increase the probability of bruxism. Therefore, the findings from this research could help the paediatrician and dentist guide the anamnesis and inform parents about the risks associated with increased levels of masticatory muscle activity in their child. However, further studies with similar sample ranges are recommended to more accurately assess the relationship between age, sex, orofacial and social factors, and bruxism.

## 5. Conclusions

The prevalence of sleep bruxism in preschool children aged 3 to 5 years was 28.9%. Factors associated with this habit include age, sex, open bite, and increased overbite and crossbite.

Early diagnosis and treatment are crucial, Considering the long-term effects of this habit on the child’s overall health status. We strongly recommend that paediatricians and other health professionals identify and inform parents about parafunctional behaviours, emphasising the multidisciplinary approach and the necessity of managing its development.

The primary and noticeable symptom of bruxism in children is grinding. The unpleasant, repetitive sound that occurs during sleep serves as a clear indication of non-functional tooth contact. It is also valuable for parents to be aware of the risk factors and factors associated with bruxism. Its multifactorial aetiology necessitates treatments that extend beyond dental care and encompass various aspects of human well-being.

## Figures and Tables

**Figure 1 healthcare-11-01450-f001:**
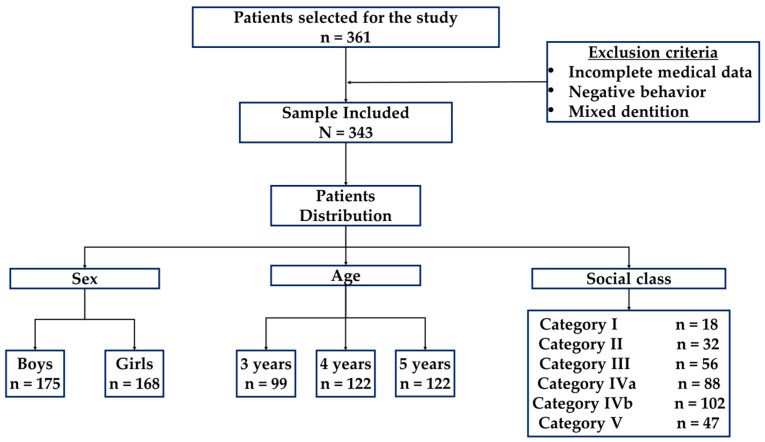
Sample distribution by sex, age, and social status.

**Figure 2 healthcare-11-01450-f002:**
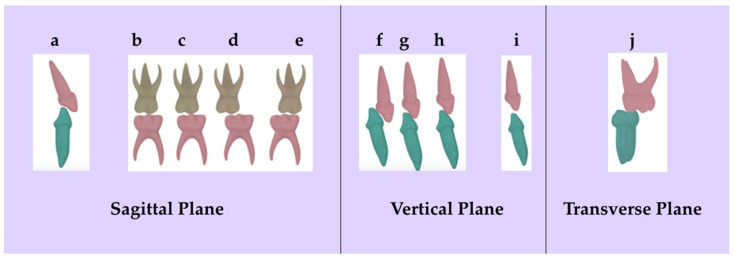
Occlusal characteristics in the three anatomical planes: (**a**) overjet, (**b**) flush terminal plane, (**c**) short mesial step, (**d**) long mesial step, (**e**) distal step, (**f**) overbite 3/3, (**g**) overbite 2/3, (**h**) overbite 1/3, (**i**) open bite, and (**j**) crossbite.

**Figure 3 healthcare-11-01450-f003:**
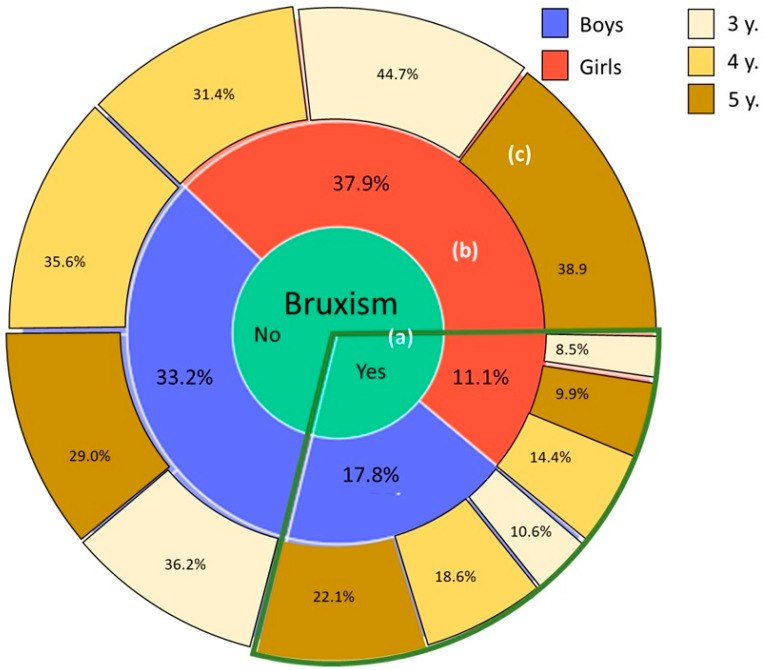
Percentages of each subgroup based on the total population recruited. The central circumference (**a**) is the presence or absence of bruxism. The middle circumference (**b**) is the distribution by sex. Additionally, the external circumference (**c**) is the distribution by age.

**Figure 4 healthcare-11-01450-f004:**
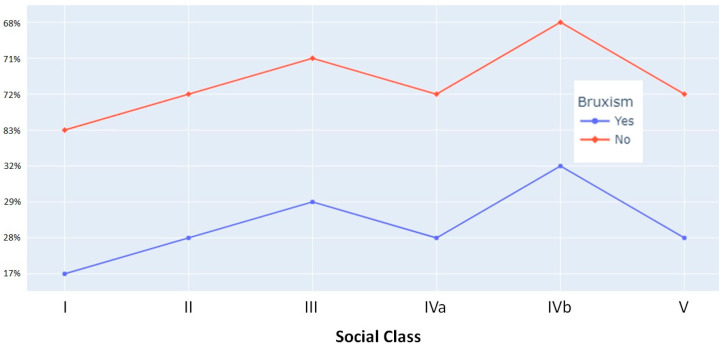
Line graph comparing the presence or absence of bruxism by family social class.

**Figure 5 healthcare-11-01450-f005:**
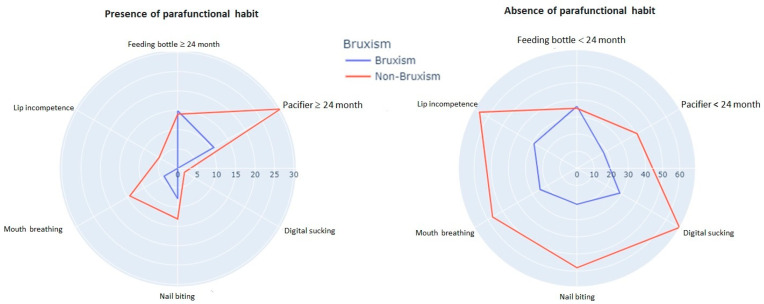
Radial plots of factors for parafunctional habits as risk factors.

**Figure 6 healthcare-11-01450-f006:**
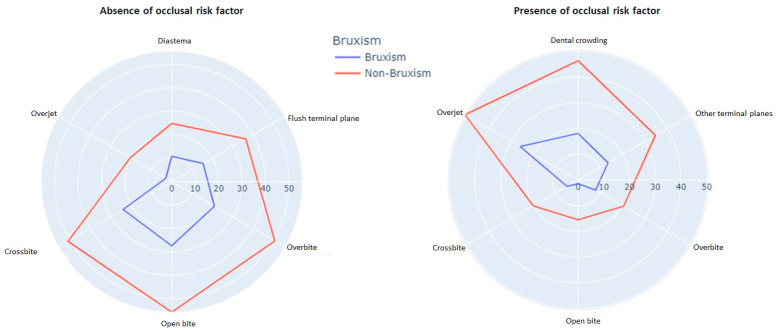
Radial plots of occlusal characteristics as risk factors.

**Table 1 healthcare-11-01450-t001:** Social class classification [25].

Social Class	
I	Directors of Public Administration and companies with ten or more employees Professions associated with second and third-cycle university degrees
II	Company managers with less than ten employees Professions associated with a first-cycle university degree. Technicians, artists, and athletes
III	Administrative employees and support professionals for administrative and financial management Workers of personal and security services Free-lancers Manual worker supervisors
IVa	Skilled manual workers
IVb	Semi-skilled manual workers
V	Unskilled workers

**Table 2 healthcare-11-01450-t002:** Association between social class and possible sleep bruxism.

Social Class	Bruxism (Yes) *n* (%)	Bruxism (No) *n* (%)	*p*
I	3 (3.1)	15 (6.1)	0.854
II	9 (9.1)	23 (9.4)	
III	16 (16.2)	40 (16.4)	
IVa	25 (25.3)	63 (25.8)	
IVb	33 (33.3)	69 (28.3)	
V	13 (13.1)	34 (13.9)	
Total	99 (100)	244 (100)	

**Table 3 healthcare-11-01450-t003:** Association between parafunctional habits and possible sleep bruxism.

Parafunctional Habits	Bruxism (Yes) *n* (%)	Bruxism (No) *n* (%)	PR	CI 95%	*p*
Pacifier					
≥24 months	37 (37.4)	105 (43.0)	1.27	0.78–2.04	0.132
<24 months	62 (62.6)	139 (57.0)			
Feeding Bottle					
≥24 months	51 (51.5)	48 (48.5)	1.03	0.64–1.64	0.907
<24 months	124 (50.8)	120 (49.2)			
Digital Sucking					
Presence	0 (0.0)	7 (2.9)			0.199
Absence	99 (100)	237 (97.1)			
Nail Biting					
Presence	27 (27.3)	45 (18.4)	1.66	0.96–2.87	0.069
Absence	72 (72.7)	199 (81.6)			
Mouth Breathing					
Presence	14 (14.1)	49 (20.1)	0.66	0.34–1.25	0.198
Absence	85 (85.9)	195 (79.9)			
Lip Incompetence					
Presence	0 (0.0)	9 (7.8)			0.003
Absence	99 (100)	225 (92.2)			

PR: prevalence ratio; CI 95%: confidence interval; *p*: significances chi-squared test.

**Table 4 healthcare-11-01450-t004:** Association between occlusal characteristics and possible sleep bruxism.

Occlusal Characteristics	Bruxism (Yes) *n* (%)	Bruxism (No) *n* (%)	PR	CI 95%	*p*
Diastemas					
Dental crowding	62 (62.6)	159 (62.5)	0.9	0.55–1.45	0.656
Diastemas	37 (37.4)	85 (34.8)			
Relationship of the primary molars					
Short mesial step	11 (11.1)	41 (16.8)	0.91	0.57–1.46	0.699
Long mesial step	3 (3.0)	10 (4.1)			
Distal step	18 (18.2)	34 (13.9)			
Mixed component	14 (14.1)	34 (13.9)			
Flush terminal plane	53 (53.5)	125 (51.2)			
Overjet					
Presence	27 (27.3)	70 (28.7)	0.93	0.55–1.57	0.792
Absence	72 (72.7)	174 (71.3)			
Open bite					
Presence	5 (5.1)	53 (21.7)	0.19	0.07–0.50	0.000
Absence	94 (94.9)	191 (78.3)			
Crossbite					
Presence	17 (17.2)	69 (28.3)	0.53	0.29–0.95	0.032
Absence	82 (82.9)	175 (71.7)			
Overbite					
1/3	26 (26.3)	59 (24.2)	3.58	1.76–7.28	0.002
2/3	34 (34.3)	57 (23.4)			
3/3	29 (29.3)	58 (23.8)			
No	10 (10.1)	70 (28.7)			

PR: prevalence ratio; CI 95%: confidence interval; *p*: significances chi-squared test.

## Data Availability

The data presented in this study are available upon request from the corresponding author (M.D.-P.).

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
