# Peer review of "Prevalence of Possible Sleep Bruxism and Its Association with Social and Orofacial Factors in Preschool Population"

_healthcare, 2023, doi:10.3390/healthcare11101450_

Round 1
Reviewer 1 Report
Interesting work, the topic of bruxism is very important when it comes to the development of the child and the later functioning of the entire dentition.
The idea for an interesting study, the study of a specific population on the island, i.e. in a fairly closed community. Could this cause an increase in the incidence of this disease?
As a reviewer, however, I found a few comments that require clarification.
Introduction
It would be good in the introduction to describe what are the studies- tests, that can confirm bruxism, the second thing is the methods of treatment. Some examples
The idea is to get as many readers as possible interested in your article so that they can cite your work. Therefore, I would suggest a greater expansion of this part of the article.
Materials and methods
Line 63
expected proportion of 30%10, what does it mean? 283/2108 it 13%?
I would suggest using a graphical abstract to show how complex your research method was. You can see an example. It's about what group of the patient was, how many were interviewed, broken down by age groups, parents from what social groups, etc. It will be clearer.
And follow with this way it will be much more transparent
Line 77
The diagnosis of bruxism was based on the information reported by the parents. - what kind of information may indicate the presence of bruxism?
How many questions were there in the survey and what did they concern, because I am also missing, how many questions were there about economic status?
In the additional information to this article, you could provide what the questionnaire was, it would make it easier for people who deal with similar topics to create a template for research...
Table 1 is hard to read. Decide which occupational groups have been assigned to the first group and which to the second group.
Line 86
Fedesa JS 500- dental chair producer, country?
Clinical exam- to this part I would add an illustrative figure showing all these axes. I am not an orthodontist, so it would be a big help for me as a reader
Results
On materials and methods, you wrote 283 patients but now you are saying 343 children. Why such difference, please explain.
Line 129
Figure 1. Percentages of each subgroup based on the total population recruited. (a) presence or absence of bruxism. (b) distribution by sex. (c) distribution by age
I can see a, but not b an c in the diagram, they are some number and % but you couldn’t explain what it is, please precise, to make this Figure much more clear and understandable.
Table 3
>24 meses- 24 months?
Missing line before and after Lip incompetence similar way as in upper part of the table?
Interesting presentation of the results in the form of pie charts, congratulations
Discussion
Lien 163
This is the only study that we have found in the scientific literature that determines the prevalence with the same age ranges as our research (5.2%, 11.6%, 11.7% look at this for example:
Bulanda, Sylwia & Ilczuk-RypuÅ‚a, Danuta & Nitecka-Buchta, Aleksandra & Nowak, Zuzanna & Baron, Stefan & Postek-StefaÅ„ska, Lidia. (2021). Sleep Bruxism in Children: Etiology, Diagnosis and Treatment—A Literature Review. International Journal of Environmental Research and Public Health. 18. 9544. 10.3390/ijerph18189544.
I still lack practical advice in your work, what parents should pay attention to, what the symptoms of bruxism may be, what they should do about it, etc. And of course, what are the treatment methods?
Good luck with further research
Minor language corrections, e.g. months are written in Spanish instead of English
Author Response
The authors appreciate your comments and suggestions. This helps improve the way we present research. Below we will elaborate and address each of your comments. We particularly appreciate your positive feedback that has encouraged us to look into this issue further.

Reviewer 2 Report
Dear authors, congratulations for the interesting topic chosen. To facilitate understanding, please specify the following:
-is the sample considered significant for the preschool population in your country?
- was crowding in the temporary dentition associated with other developmental anomalies?
- was the degree of anxiety of the children in relation to the examination/dental treatments considered?
Thank you!
Author Response

(The authors gave the same response as above.)

Reviewer 3 Report
The authors analyzed the prevalence of sleep bruxism in the preschool teenage population (3-5 years of age). The study compared the association of bruxism with social and orofacial factors in early teenagers.
From the study, the authors observed:
· About 30% of the population presented bruxism, and gender and sex were significant factors.
· The social class of the parents had no significant effect on bruxism.
· Lip competence, open bite, crossbite, and overbite had a significant effect on bruxism.
The authors did a great job describing the study results using tables and figures. In addition, the authors discussed their results and compared them with the previous results in the literature in detail. Great work!
There are some minor issues if addressed, can improve the article:
Abstract:
a) Line 18-19, “Of the total sample… sleep bruxism (28.5%).. at the age of 5 years (41.4%).” The percentages mentioned do not match those from the results section, lines 120-125. The sentence states that bruxism decreased in males, but males had higher bruxism (61.6%) than females.
b) Also, the same terminology, “boys and girls” or “males and females” could be used across the entire article to make it consistent.
Materials and Methods:
a) Line 62-64 is confusing as the expected proportion of 30% is for what? Also, the total number of patients in the abstract, these lines, and the results section (Lines 118-120) differ. Why so?
Results:
a) The total number of patients in the study (343) mentioned in lines 118-120 differs from those in the abstract, materials, and methods.
b) Lines 119-120 states that 18 participants were excluded from the study. The total number and proportion of participants for each category still comprise 343 teenagers.
Table 3:
a) It might be better to replace “messes” in the first column with “months” for English readers, just like in Figure 3.
Conclusion:
a) The percentage “28.5%” for sleep bruxism in preschool teenagers differs from what is reported in the results section of the article.
Author Response

(The authors gave the same response as above.)
